# Achieving Rotational Invariance with Bessel-Convolutional Neural Networks

**Valentin Delchevalerie,**[1] **Adrien Bibal,**[2] **Benoît Frénay,**[2]* **Alexandre Mayer**[3]*

[1] PReCISE, NADI & naXys institutes, University of Namur, Belgium
[2] PReCISE, NADI institute, University of Namur, Belgium
[3] Department of Physics, naXys institute, University of Namur, Belgium
{valentin.delchevalerie, adrien.bibal, benoit.frenay, alexandre.mayer}@unamur.be

## Abstract

For many applications in image analysis, learning models that are invariant to translations and rotations is paramount. This is the case, for example, in medical imaging where the objects of interest can appear at arbitrary positions, with arbitrary orientations. As of today, Convolutional Neural Networks (CNN) are one of the most powerful tools for image analysis. They achieve, thanks to convolutions, an invariance with respect to translations. In this work, we present a new type of convolutional layer that takes advantage of Bessel functions, well known in physics, to build Bessel-CNNs (B-CNNs) that are invariant to all the continuous set of possible rotation angles by design.

## 1   Introduction

Deep learning models, and more particularly Convolutional Neural Networks (CNNs), are known as being among the most powerful tools for image analysis. For this reason, they are still constantly upgraded in order to achieve better performance [1]. One of the main reasons why CNNs are so much used in computer vision lies in the fact that they achieve translation invariance thanks to convolutions. Filters sweep the image locally and patterns can be recognized regardless of their absolute position in the image. However, some other important types of invariance are more difficult to obtain. It is for example the case for the rotational invariance, which is relevant for many applications. One could for example consider medical imaging where tissues, cells, tumors or other objects of interest have a local, arbitrary, orientation in the images [2]. Another example is satellite imaging of ships, where both the global orientation of the satellite and the local orientation of a ship are arbitrary [3].

Multiple works proposed solutions in order to bring rotational invariance in CNNs. However, many of them (i) only make the model more robust to rotations without providing guarantees regarding the rotational invariance [4, 5, 6, 7, 8], (ii) only provide guarantees for a finite set of rotation angles [9, 10, 11, 12, 13, 14, 15] or (iii) only handle global rotational invariance, while a local one is sometimes more relevant [4, 5, 6]. In addition to that, and more similarly to the method proposed in this paper, other works present solutions to build CNNs that are rotational invariant for all rotation angles, while providing mathematical guarantees. One can for example cite [16, 17, 18]. Our work is in the continuity of those. We integrate a new kind of convolutional layer in CNNs to build Bessel-Convolutional Neural Networks (B-CNNs). In B-CNNs, Bessel functions from physics are used to build a representation of the images that is more adapted to deal with rotations. We propose a new way to use this representation in order to compute feature maps in a rotational equivariant way (see Figure 1 for an example). This rotational equivariance can then lead to rotational invariance with a proper choice of architecture.

---

*B. Frénay and A. Mayer are co-last authors.

35th Conference on Neural Information Processing Systems (NeurIPS 2021).

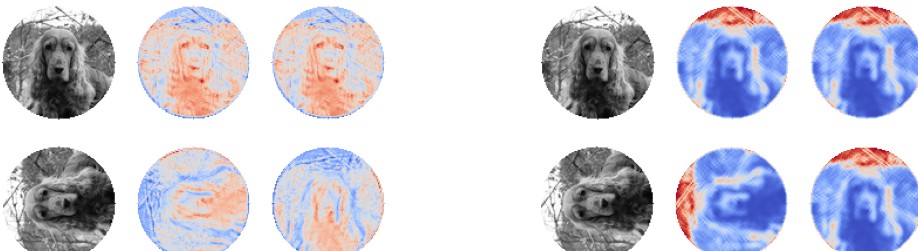

Figure 1: This figure illustrates the lack of rotational equivariance in standard CNNs (left), as opposed to the rotational equivariance of B-CNNs (right). For each triplet of images, the first image is the input image, the second is the feature map obtained, and the last one is the feature map reoriented to make the comparison easier. It can be seen that the last images in the two left triplets for the standard CNN are different. Hence, a different orientation of the input image produces a different feature map. However, feature maps are identical except for a rotation for the B-CNN on the right.

Section 2 formally defines the problem of rotational invariance in CNNs and provides the necessary notations. Next, Section 3 presents the related works on bringing rotational invariance into CNNs. Our method, B-CNN, is then introduced in Section 4, along with some background on Bessel functions. Experiments are described and discussed in Section 5. Finally, a more general discussion about some aspects of B-CNNs, as well as some future works, are presented in Section 6 before concluding the paper in Section 7.

## 2 Background and problem definition

For several applications in computer vision, patterns of interest may appear in arbitrary orientations. This typically happens in medical imaging [2], satellite imaging [3], astronomical imaging [19], texture recognition [20], etc. In such cases, the fact that the orientation is meaningless can be used as a prior during learning through a feature extraction that is orientation-agnostic. However, building deep learning models that present guarantees with respect to the rotational invariance is not trivial. Indeed, in the particular case of Convolutional Neural Networks (CNNs), the computation of a feature map is clearly not rotation invariant since, in general, for a rotation of an angle $\alpha$, we have

$$\sum_{m=0}^{k_1-1}\sum_{n=0}^{k_2-1} I\left(x-m,y-n\right)K\left(m,n\right) \neq \sum_{m=0}^{k_1-1}\sum_{n=0}^{k_2-1} I\left(x'-m,y'-n\right)K\left(m,n\right), \qquad (1)$$

where $x' = x\cos\alpha - y\sin\alpha$, $y' = x\sin\alpha + y\cos\alpha$, $I$ is the input of the convolutional layer and $K$ represents a particular filter of size $k_1 \times k_2$. Figure 1 illustrates this lack of invariance.

Two important notions are central in this work: rotational invariance and rotational equivariance. On the one hand, if $f\left(I\right)$ represents the computation of feature maps in a particular layer of a CNN, $f\left(I\right)$ is rotational invariant if $f\left(R\left(\alpha\right)I\right) = f\left(I\right), \forall\alpha \in [0, 2\pi)$, where $R\left(\alpha\right)$ is an operator that rotates $I$ by an angle $\alpha$. However, if $f\left(R\left(\alpha\right)I\right) = R\left(\alpha\right)f\left(I\right), \forall\alpha \in [0, 2\pi)$, then $f\left(I\right)$ is rotational equivariant since feature maps are identical except for a rotation (see Figure 1). For particular architectures where the feature maps in the final layer have a size of $1 \times 1$, this rotational equivariance leads to a rotational invariance. Indeed, if $I$ is one single value, $R\left(\alpha\right)I = I$. Such a strong rotational invariance is something usually not guaranteed, or sometimes only for a finite subset of $[0, 2\pi)$.

## 3 Related work on rotational invariance for CNNs

The use of Bessel functions in the Machine Learning community has already been proposed for image denoising, classification or recognition purposes. One can for example cite [21, 22, 23, 24], which already use a decomposition using Bessel functions similar to the one used in this paper. However, these works differ from ours since their objective is to build a rotational invariant covariance matrix, and to obtain rotational invariant models on top of this, while the objective of B-CNNs is to perform a rotational invariant convolutional operation between filters and images.

Methods for introducing rotational invariance in CNNs can be categorized in different groups: those (i) that just make CNNs more robust to rotations, without providing any mathematical guarantees, those (ii) that provide mathematical guarantees but only for a finite set of rotations (typically obtained by a discrete group of symmetries), and finally those (iii) that introduce a continuous rotational invariance while providing mathematical guarantees.

Among the first category of methods, one can cite data augmentation [4], spatial transformer networks (STN) [5], rotation-invariant and fisher discriminative CNNs (RFID-CNNs) [6], deformable CNNs [7], and SIFT-CNNs [8]. For the second category, one of the best-known method is Group CNNs (G-CNNs) [9], which define the transformations for which an invariance is desired using the theory of groups and symmetries. However, one can also cite transformation-invariant pooling (TI-POOLING) [10], deep symmetry networks (DSN) [11], steerable CNNs [12], steerable filter CNNs (SFCNNs) [13], dense steerable filter CNNs [14], and spherical CNNs [15]. Finally, there are already a few methods in the third category, i.e. methods that are invariant to the continuous set of rotations. General E(2)-equivariant steerable CNNs [17] present a general solution to obtain E(2)-equivariant CNNs by constraining the kernels using a group representation, LieConv [3] generalizes equivariance to arbitrary transformations from Lie groups, and Harmonic networks [16] use circular harmonics to achieve a rotational invariance by maintaining a disentanglement of rotation orders in the network. Our work is in the continuity of these techniques. It proposes a new framework to achieve the invariance. For example, compared to harmonic networks, B-CNNs present different advantages: (i) it involves real-valued feature maps, which make B-CNNs easier to use with already existing techniques (it can use classic non-linear activation functions, classic batch-normalization, etc.), (ii) it uses an arbitrary number of rotation orders, and (iii) it can also be made invariant to other types of invariance in future works.

In the next section, we introduce Bessel-CNNs, based on the use of the Bessel functions well known in physics, to make CNNs invariant by design to all rotation angles in the continuous set $[0, 2\pi)$.

## 4 Exploiting Bessel functions for CNNs

The mathematical background on Bessel functions and the motivation of their use for rotational invariant image recognition tasks are presented in this section. Some of these developments are inspired from the use of Bessel functions in physics [25]. These mathematical developments lead to a new type of Convolutional Neural Networks (CNN) that we call Bessel-Convolutional Neural Network (B-CNN). Mathematical evidence of a rigorous rotational invariance is pointed out.

### 4.1 Bessel functions

Bessel functions are particularly well known in physics because they arise when solving some important problems in polar or cylindrical coordinates. For example, they appear when dealing with wave or heat propagation. Bessel functions may be of different order $\nu$ and are defined as a particular solution of the Bessel's differential equation

$$x^2 \frac{d^2 y}{dx^2} + x \frac{dy}{dx} + \left(x^2 - \nu^2\right) y = 0. \tag{2}$$

This equation involves two particular kinds of Bessel functions $J_\nu(x)$ and $Y_\nu(x)$, called the Bessel function of the first and of the second kind, respectively. In this work, $\nu$ must be an integer and we only consider $J_\nu(x)$ since $Y_\nu(x)$ diverges for $x = 0$. For $\nu \in \mathbb{N}$, $J_\nu(x)$ and $J_{-\nu}(x)$ are not independent and

$$J_{-\nu}(x) = (-1)^\nu J_\nu(x). \tag{3}$$

Bessel functions also satisfy

$$J_\nu(-x) = (-1)^\nu J_\nu(x), \tag{4}$$

which means that $J_\nu$ is an even function if $\nu$ is even, and an odd function otherwise. Bessel functions of the first kind can be used to build a particular basis for the representation in polar coordinates of images defined in a circular domain of radius $R$, as

$$\left\{ N_{\nu,j} J_\nu(k_{\nu,j}\rho) e^{i\nu\theta}, \forall \nu, j \in \mathbb{N} \right\}, \text{ where } N_{\nu,j} = 1 \Big/ \sqrt{2\pi \int_0^R \rho J_\nu^2(k_{\nu,j}\rho) \, d\rho}, \tag{5}$$

forms an orthonormal basis for all squared-integrable functions $f$ such that $f : D^2 \subset \mathbb{R}^2 \longrightarrow \mathbb{R}$ (where the domain $D^2$ is a disk in $\mathbb{R}^2$). Indeed, we can show that

$$\int_0^{2\pi} \int_0^R \rho \left[ N_{\nu,j} J_\nu \left( k_{\nu,j}\rho \right) e^{i\nu\theta} \right]^* \left[ N_{\nu',j'} J_{\nu'} \left( k_{\nu',j'}\rho \right) e^{i\nu'\theta} \right] d\theta d\rho = \delta_{\nu\nu'} \delta_{jj'}, \tag{6}$$

if the $k_{\nu,j}$ are solutions of either $J_\nu \left( k_{\nu,j} R \right) = 0$ or $J'_\nu \left( k_{\nu,j} R \right) = 0$ [26]. We use $J'_\nu \left( k_{\nu,j} R \right) = 0$ as condition for the $k_{\nu,j}$ because it makes it more convenient to represent arbitrary functions (by the fact that for $\nu = 0$ there is a $k_{\nu,j} = 0$ with this condition, we can use $J_0(0.\rho)e^{i.0.\theta} = 1$ to describe an arbitrary constant intensity) [25]. The key advantage of using this basis is that it can deal easily with rotations. Later, this basis will be used to build a new representation for images.

## 4.2 Bessel coefficients

An arbitrary function $\Psi \left( \rho, \theta \right) \in \mathbb{R}$, where $\rho$ and $\theta$ are polar coordinates, can be expressed with the basis in Equation (5) as

$$\Psi \left( \rho, \theta \right) = \sum_{\nu=-\infty}^{\infty} \sum_{j=0}^{\infty} \Psi_{\nu,j} \, N_{\nu,j} J_\nu \left( k_{\nu,j}\rho \right) e^{i\nu\theta}, \tag{7}$$

where the Bessel coefficients $\Psi_{\nu,j}$ are obtained by projecting $\Psi \left( \rho, \theta \right)$ on this particular basis

$$\Psi_{\nu,j} = \int_0^{2\pi} \int_0^R \rho \left[ N_{\nu,j} J_\nu \left( k_{\nu,j}\rho \right) e^{i\nu\theta} \right]^* \Psi \left( \rho, \theta \right) d\theta d\rho. \tag{8}$$

In order to be mathematically exact when describing $\Psi(\rho,\theta)$ using its Bessel coefficients, one should consider all $\nu \in \{-\infty, ..., \infty\}$ and all $j \in \{0, ..., \infty\}$ to represent $\Psi \left( \rho, \theta \right)$ faithfully. Nevertheless, as only a finite set of coefficients can be considered in practice, values of $\nu_{max}$ and $j_{max}$ need to be chosen. By taking into account Properties (3) and (4) when looking at Equation (8) and as $\Psi \left( \rho, \theta \right) \in \mathbb{R}$, we end up with the relations

$$\begin{cases} \Re \left( \Psi_{-\nu,j} \right) = (-1)^\nu \, \Re \left( \Psi_{\nu,j} \right) \\ \Im \left( \Psi_{-\nu,j} \right) = (-1)^{\nu+1} \, \Im \left( \Psi_{\nu,j} \right), \end{cases} \tag{9}$$

where $\Re$ and $\Im$ stands for the real and imaginary part, respectively. One can then only compute the Bessel coefficients for $\nu$ (resp $j$) in $\{0, ..., \nu_{max}$ (resp. $j_{max})\}$ since $\Psi_{-\nu,j}$ and $\Psi_{\nu,j}$ are not independent. Also, note that, when $\nu$ or $j$ increases, $k_{\nu,j}$ also increases. The values of $\nu_{max}$ and $j_{max}$ can then be determined through a mathematical insight about the maximal value of $k_{\nu,j}$ to use in a given problem (for example, using the Nyquist frequency, as in [21]).

To understand now why working with Bessel coefficients is particularly useful, one can see how an arbitrary rotation of $\Psi \left( \rho, \theta \right)$ by an angle $\alpha$ modifies its Bessel coefficients $\Psi_{\nu,j}$. To do so, let us consider $\Psi^{rot} \left( \rho, \theta \right) = \Psi \left( \rho, \theta - \alpha \right)$ for an angle $\alpha \in [0, 2\pi)$. Its Bessel coefficients are given by

$$\Psi_{\nu,j}^{rot} = \int_0^{2\pi} d\theta \int_0^R d\rho \rho \left[ N_{\nu,j} J_\nu \left( k_{\nu,j}\rho \right) e^{i\nu\theta} \right]^* \Psi^{rot} \left( \rho, \theta \right). \tag{10}$$

By defining $\theta' = \theta - \alpha$, this leads to

$$\Psi_{\nu,j}^{rot} = \int_0^{2\pi} d\theta' \int_0^R d\rho \rho \left[ N_{\nu,j} J_\nu \left( k_{\nu,j}\rho \right) e^{i\nu\theta'} \right]^* \Psi \left( \rho, \theta' \right) e^{-i\nu\alpha} = \Psi_{\nu,j} e^{-i\nu\alpha}. \tag{11}$$

This motivates our work, as a rotation of an arbitrary function by an angle $\alpha$ only modifies its Bessel coefficients by a multiplying factor $e^{-i\nu\alpha}$, which makes rotations conveniently expressed in the Fourier-Bessel transform domain (analogously to how the Fourier transform maps translations to multiplications by complex exponentials). Therefore, if $\Psi \left( \rho, \theta \right)$ represents an image, its Bessel coefficients constitute a more adapted representation to build rotational invariant operations.

## 4.3 Defining a rotational invariant operation

In order to build a rotational invariant operation $\bullet$ between the Bessel coefficients $\Psi_{\nu,j}$ and a filter made of the same number of complex numbers $K_{\nu,j}$, one can consider

$$\mathbf{K} \bullet \mathbf{\Psi} = \frac{1}{2\pi} \int_0^{2\pi} | \sum_{\nu,j} K_{\nu,j}^* \Psi_{\nu,j} e^{-i\nu\alpha} |^2 d\alpha. \tag{12}$$

This operation is, by design, necessarily invariant to rotations as Equation (11) shows that multiplying each $\Psi_{\nu,j}$ by $e^{-i\nu\alpha}$ is equivalent to applying a rotation to the image. By performing the integration for $\alpha$ going from 0 to $2\pi$, the image achieves a complete rotation around itself. Therefore, $\mathbf{K} \bullet \mathbf{\Psi}$ does not depend anymore on the particular initial orientation of the image. Note that, without the squared modulus, we can show that only the coefficients $\Psi_{0,j}$ contribute to the final results. The operation would therefore rely on an incomplete representation of the image.

Using Equation 12 as such may look like a computationally expensive brute force strategy, since it requires to evaluate the numerical integration over $\alpha$. However, given that

$$|\sum_{i=1}^{k} \alpha_i z_i|^2 = \sum_{m,j} \Re\left(\alpha_m\right) \Re\left(\alpha_j\right) |z_m z_j| \cos\left(\theta_m - \theta_j\right) + \sum_{m,j} \Im\left(\alpha_m\right) \Im\left(\alpha_j\right) |z_m z_j| \cos\left(\theta_m - \theta_j\right)$$
$$- 2\sum_{m,j} \Re\left(\alpha_j\right) \Im\left(\alpha_m\right) |z_m z_j| \sin\left(\theta_m - \theta_j\right), \tag{13}$$

where $\alpha_i$ and $z_i = |z_i|e^{i\theta_i}$ are complex numbers, the squared modulus in Equation (12) becomes

$$\sum_{\substack{\nu,j \\ \nu',j'}} \Re\left(K_{\nu,j}^*\right) \Re\left(K_{\nu',j'}^*\right) |\Psi_{\nu,j}\Psi_{\nu',j'}| \cos\left(\theta_{\nu,j} - \theta_{\nu',j'} - \alpha\left(\nu - \nu'\right)\right)$$
$$+ \sum_{\substack{\nu,j \\ \nu',j'}} \Im\left(K_{\nu,j}^*\right) \Im\left(K_{\nu',j'}^*\right) |\Psi_{\nu,j}\Psi_{\nu',j'}| \cos\left(\theta_{\nu,j} - \theta_{\nu',j'} - \alpha\left(\nu - \nu'\right)\right)$$
$$- 2\sum_{\substack{\nu,j \\ \nu',j'}} \Im\left(K_{\nu,j}^*\right) \Re\left(K_{\nu',j'}^*\right) |\Psi_{\nu,j}\Psi_{\nu',j'}| \sin\left(\theta_{\nu,j} - \theta_{\nu',j'} - \alpha\left(\nu - \nu'\right)\right), \tag{14}$$

if $\Psi_{\nu,j} = |\Psi_{\nu,j}|e^{i\theta_{\nu,j}}$. The three terms should be integrated over $\alpha$ thanks to Equation (12), but only the trigonometric functions are $\alpha$-dependent. A simple integration leads to

$$\int_0^{2\pi} \text{sc}\left(\theta_{\nu,j} - \theta_{\nu',j'} - \alpha\left(\nu - \nu'\right)\right) d\alpha = \begin{cases} 2\pi \, \text{sc}\left(\theta_{\nu,j} - \theta_{\nu',j'}\right) \textbf{ if } \nu = \nu' \\ 0 \text{ otherwise,} \end{cases} \tag{15}$$

where sc can represent the cosine or the sine function. Therefore,

$$\mathbf{K} \bullet \mathbf{\Psi} = \sum_{\nu,j,j'} \Re\left(K_{\nu,j}^*\right) \Re\left(K_{\nu,j'}^*\right) |\Psi_{\nu,j}\Psi_{\nu,j'}| \cos\left(\theta_{\nu,j} - \theta_{\nu',j'}\right)$$
$$+ \sum_{\nu,j,j'} \Im\left(K_{\nu,j}^*\right) \Im\left(K_{\nu,j'}^*\right) |\Psi_{\nu,j}\Psi_{\nu,j'}| \cos\left(\theta_{\nu,j} - \theta_{\nu',j'}\right)$$
$$- 2\sum_{\nu,j,j'} \Im\left(K_{\nu,j}^*\right) \Re\left(K_{\nu,j'}^*\right) |\Psi_{\nu,j}\Psi_{\nu,j'}| \sin\left(\theta_{\nu,j} - \theta_{\nu',j'}\right), \tag{16}$$

and by using once again the result presented in Equation (13), we have

$$\mathbf{K} \bullet \mathbf{\Psi} = \sum_{\nu} |\sum_{j} K_{\nu,j}^* \Psi_{\nu,j}|^2, \tag{17}$$

which is therefore equivalent to Equation (12), but much more convenient. By construction, the results obtained by Equation (17) do not depend on the particular orientation of the image. Furthermore, $\mathbf{K} \bullet \mathbf{\Psi} \in \mathbb{R}$ (with $K_{\nu,j} \in \mathbb{C}$ and $\Psi_{\nu,j} \in \mathbb{C}$), which is something convenient in order to use, for example, classic activation functions. Using Equation 12 (or Equation 17) to achieve the rotational invariance is one of the main contribution of this paper.

### 4.4 Bessel-Convolutional Neural Networks (B-CNNs)

In standard CNNs, a feature map is the result of the convolution between the image and a filter $\mathbf{K}$ of $k_1 \times k_2$ weights, as described in Equation (1). The weights of this filter are real numbers that are tuned during the learning process of the network. In Bessel-Convolutional Neural Networks (B-CNNs), the process to build feature maps is slightly different as the representation of the image based on Bessel

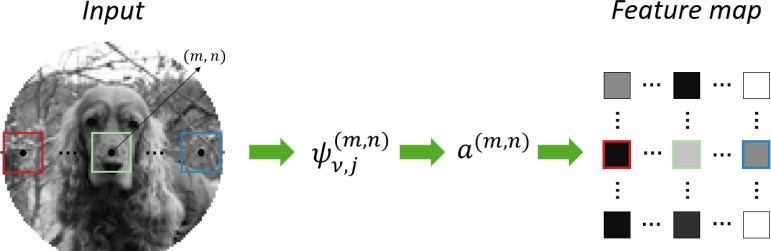

Figure 2: Illustration of how the feature maps are computed in B-CNNs.

coefficients are used. A filter $\mathbf{K}$ made of $(\nu_{max} + 1) \times (j_{max} + 1)$ complex weights $K_{\nu,j}$ is used to sweep the image over sub-regions of size $k \times k$. Note that $\nu_{max}$ and $j_{max}$ are generally larger than $k_1$ and $k_2$, which results in an increase of the number of trainable parameters in each layer. However, as it is easier for B-CNNs to deal with rotational invariance, we show in Section 5 that architectures with less layers than for other methods still achieve similar or even better performances. For each sub-region of the image centered at position $(m, n)$, its Bessel coefficients $\Psi_{\nu,j}^{(m,n)}$ are computed, and

$$a^{(m,n)} = \sum_\nu | \sum_j K_{\nu,j}^* \Psi_{\nu,j}^{(m,n)} |^2 \tag{18}$$

is computed, where $a^{(m,n)}$ is a real value that is invariant to rotation, i.e., it does not depend on the particular orientation of the sub-region. All the values obtained for the different sub-regions constitute the feature map, which is then rotation equivariant. This is schematized in Figure 2.

### 4.5 An efficient implementation of B-CNNs

Implementing B-CNNs as described in Section 4.4 can be slow, as it requires to evaluate Equation (8) for each sub-region of the image. However, developing Equation (18) with Equation (8) gives

$$a^{(m,n)} = \sum_\nu | \sum_j K_{\nu,j}^* \int_0^{2\pi} \int_0^R N_{\nu,j} J_\nu (k_{\nu,j}\rho) \, e^{-i\nu\theta} \Psi^{(m,n)} (\rho,\theta) \, \rho d\rho d\theta |^2$$

$$= \sum_\nu | \int_0^{2\pi} \int_0^R \Psi^{(m,n)} (\rho,\theta) \sum_j K_{\nu,j}^* N_{\nu,j} J_\nu (k_{\nu,j}\rho) \, e^{-i\nu\theta} \rho d\rho d\theta |^2, \tag{19}$$

where $\Psi^{(m,n)} (\rho,\theta)$ represents a particular sub-region of the image centered at $(m, n)$ in polar coordinates. When going back to a Cartesian coordinates system, one can get

$$a^{(m,n)} = \sum_\nu | \int_{-\frac{R}{2}}^{\frac{R}{2}} \int_{-\frac{R}{2}}^{\frac{R}{2}} I (m - x, n - y) \sum_j K_{\nu,j}^* N_{\nu,j} \widetilde{J}_\nu \left( k_{\nu,j} \sqrt{x^2 + y^2} \right) e^{-i\nu\widetilde{\theta}(x,y)} dxdy |^2, \tag{20}$$

where $I (x, y)$ is the input of the layer, $\widetilde{\theta} (x, y) = \pi + \arctan \frac{y}{x}$ and $\widetilde{J}_\nu (k_{\nu,j}\rho)$ is defined as

$$\widetilde{J}_\nu (k_{\nu,j}\rho) = \begin{cases} J_\nu (k_{\nu,j}\rho) \text{ if } \rho \leq R \\ 0 \text{ otherwise.} \end{cases} \tag{21}$$

Finally, by defining $T_{\nu,j} (x, y) = N_{\nu,j} \widetilde{J}_\nu \left( k_{\nu,j} \sqrt{x^2 + y^2} \right) e^{-i\nu\widetilde{\theta}(x,y)}$, Equation (20) leads to

$$\mathbf{a} = \sum_\nu |I (x, y) * \sum_j K_{\nu,j}^* T_{\nu,j} (x, y) |^2 = \sum_\nu |I (x, y) * F_\nu (x, y) |^2, \tag{22}$$

where $*$ denotes a convolutional product and $\mathbf{F}_\nu (x, y)$ corresponds to a filter modified by the Bessel functions. Compared to a standard CNN, the only difference is that, actually, a layer in B-CNN performs $\nu_{max} + 1$ convolutions instead of one, and it also needs to update the filters with the Bessel functions. Nevertheless, $T_{\nu,j} (x, y)$ is part of the model and only needs to be computed once, as it does not depend on the input of the layer, but only on the choice of $\nu_{max}$ and $j_{max}$.

# 5 Experiments

This section presents the details of the experiments used to test the rotational invariance of B-CNNs, including the datasets (Section 5.1), the baseline architectures (Section 5.2) and the experimental setup (Section 5.3). Experimental results are presented in Section 5.4 and a discussion on the rotational equivariance of feature maps is proposed in Section 5.5.

## 5.1 Datasets

In this comparative study, three datasets are used in an increasing complexity. In all of them, orientations present in the test set are not present in the training set. The first dataset is MNIST [27], a classic baseline of $28 \times 28$ images of handwritten digits, where images in the test set are randomly rotated by an angle in $[0, 2\pi)$. For all runs, the training set contains $60,000$ images and the test set $10,000$ images. The second dataset is Outex-TC-00010-r dataset [28], a classic dataset for a more complex classification involving both global and local rotations. It contains $128 \times 128$ grayscale images of 24 particular textures. The training set contains 480 images with 20 orientations, and the test set is composed of 3840 images with 160 orientations. The third dataset is made of $128 \times 128 \times 3$ brain MRI images [29] that are either cancerous or non-cancerous. This dataset is used as a real-world, more complex dataset, which contains local entities (the brain tumors) that can occur at arbitrary orientations. Each image is cropped to make the brain fill the entire image. The training and test sets contain 190 and 63 images, respectively.

## 5.2 Baseline architectures

Experiments aim to evaluate how good B-CNNs are for problems where rotational invariance is desired. We decided to use G-CNNs, as well as standard CNNs in our comparative study. For G-CNNs, the symmetry group C4 is used (rotations of angle $\frac{\pi}{2}$, $\pi$, $\frac{3\pi}{2}$ and $2\pi$). For each dataset, one architecture based on CNNs, G-CNNs and B-CNNs has been chosen by picking the one that performs the best on a validation set. Importance is given to the fact that the three models contain roughly the same number of trainable parameters, in order to perform fair comparisons. Attention is also paid to the fact that the feature map in the last convolution layer should be of size $1 \times 1$, in order to achieve rotational invariance for G-CNNs and B-CNNs.

The standard CNN architecture for MNIST is made of 7 conv. layers with 32 filters for the first two layers, and 64 filters of size $3 \times 3$ for the other layers; one max-pooling layer that operates over $2 \times 2$ regions after the second and last layers; and one dense layer with 10 units. The G-CNN architecture is similar, except that the number of filters is divided by 2 in order to keep the same number of parameters. The B-CNN for MNIST is made of 5 Bessel-convolutional (B-Conv) layers (one of 16 filters with $\nu_{max} = 11$, $j_{max} = 9$ and $k = 3$; three of 16 filters with $\nu_{max} = 9$, $j_{max} = 7$ and $k = 5$; and one of 32 filters with $\nu_{max} = 7$, $j_{max} = 5$ and $k = 7$), a max-pooling layer over $2 \times 2$ regions after the fourth layer and a dense layer with 10 units. Activation functions are relu, except for the B-CNN where tanh leads to better performances on the validation set. Adam optimization with a learning rate of $0.0001$ and an exponential decay rate of $0.9$ is used for the three models.

The standard CNN for Outex is made of 5 conv. layers (one with 48 filters of size $5 \times 5$, one with 64 filters of size $5 \times 5$, two with 64 filters of size $3 \times 3$ and one with 128 filters of size $3 \times 3$), one max-pooling layer over $3 \times 3$ regions after the first layer, one over $2 \times 2$ regions after the second and third layers, and one dense layer with 24 units. The G-CNN architecture for Outex is similar, but the number of filters is divided by 2 again to have the same number of parameters. The model based on B-CNN is made of 3 B-Conv layers (one with 16 filters with $\nu_{max} = 15$, $j_{max} = 11$ and $k = 15$; one with 32 filters with $\nu_{max} = 10$, $j_{max} = 9$ and $k = 9$; and one with 32 filters with $\nu_{max} = 7$, $j_{max} = 5$ and $k = 7$), a max-pooling layer over $3 \times 3$ regions after the first layer, one over $2 \times 2$ regions after the two others, and a dense layer with 24 units. Again, activation functions are relu, except for B-CNN where tanh are used. Adam optimization with a learning rate of $0.0005$ and an exponential decay rate of $0.9$ is used for the three models.

The standard CNN for brain MRI is made of 5 conv. layers (one with 32 filters of size $7 \times 7$, one with 32 filters of size $5 \times 5$, two with 64 filters of size $5 \times 5$ and one with 64 filters of size $3 \times 3$), one max-pooling layer over $3 \times 3$ regions after the first layer, one over $2 \times 2$ regions after the second and third layers, and one dense layer with 2 units. The G-CNN architecture is similar but the number

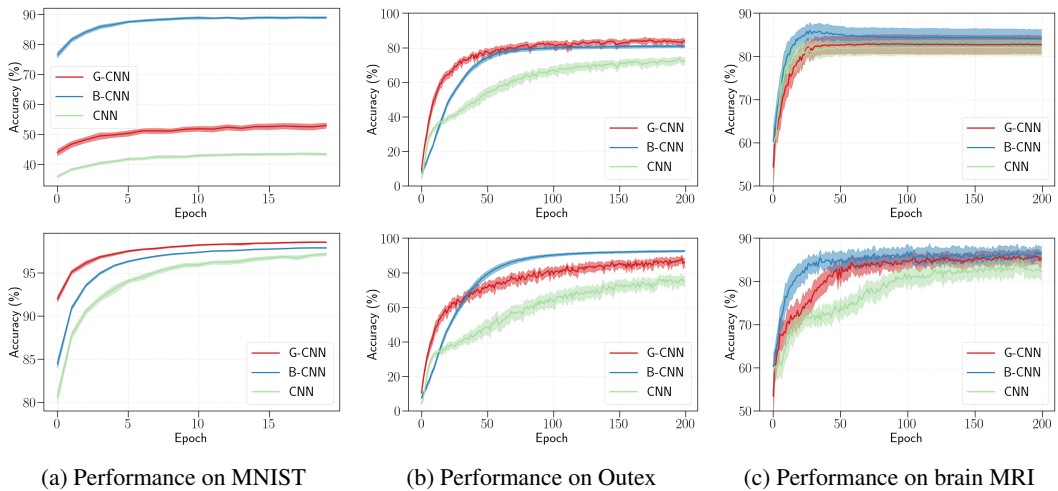

| (a) Performance on MNIST | (b) Performance on Outex | (c) Performance on brain MRI |

Figure 3: Mean test accuracy over 40 runs with $99\%$ confidence intervals for B-CNN, G-CNN and standard CNN on MNIST, Outex and brain MRI. In the first row, the training sets contain input images as they are originally in the dataset, whereas data augmentation is used in the second row.

of filters is once again divided by 2. The model based on B-CNN is made of 4 B-Conv layers (one with 32 filters with $\nu_{max} = 11$, $j_{max} = 11$ and $k = 9$; one with 16 filters with $\nu_{max} = 9$, $j_{max} = 9$ and $k = 7$; one with 16 filters with $\nu_{max} = 7$, $j_{max} = 7$ and $k = 5$; and one with 32 filters with $\nu_{max} = 7$, $j_{max} = 7$ and $k = 5$), a max-pooling layer over $3 \times 3$ regions after the first layer, one over $2 \times 2$ regions after the three other layers, and a dense layer with 2 units. Again, activation functions are relu, except for B-CNN where tanh are used. Adam optimization with a learning rate of $0.001$ and a exponential decay rate of $0.9$ is used for the three models.

## 5.3 Experimental setup

Each experiment consists of $40$ independent runs and mean accuracy scores with $99\%$ confidence intervals are reported. In order to isolate intrinsic rotational invariance, each model is trained for each dataset in two different settings. In the first setting, images are not rotated at all during training. In the second setting, input images are rotated by a random angle during training (a form of data augmentation). In order to avoid border effects due to the appearance of black corners when images are rotated for the Outex dataset, images are cropped from an initial size of $128 \times 128$ to $88 \times 88$. This is not performed for MNIST and brain MRI, since image corners are already black.

In order to highlight the ability of B-CNNs to use the invariance for the continuous set of angles as a prior knowledge, another experiment in a low data setting is also performed. This experiment is exactly the same as the previous one on the Outex dataset (with rotation of the input images), except that 75% of the training set is removed, and placed in the test set.

## 5.4 Experimental results

Figure 3 shows the mean test accuracy of B-CNNs, G-CNNs and standard CNNs on MNIST, Outex and brain MRI. The first observation is that standard CNNs have poor performance in almost all cases, which justifies the need for better methods. A second observation is that using data augmentation (second row), as opposed to using the input images as they originally are in the dataset (first row), increases the performance of all methods in all cases. Furthermore, in the setup without data augmentation, it can be observed that B-CNNs largely outperform G-CNNs and standard CNNs on MNIST. This indicates that G-CNNs need data augmentation in addition to the groups of symmetry to learn rotational invariance, while Bessel convolutions intrinsically impose the invariance.

The results are interestingly reversed for Outex, where B-CNNs perform better than G-CNNs with data augmentation, while being roughly equivalent without data augmentation. The good performance of G-CNNs without data augmentation on Outex may be explained by the fact that the texture images

contain an intrinsic data augmentation (a rotation of an image can be present in the original dataset). This is not the case of MNIST, where all images in the original dataset have the same orientation.

It can also be observed that the variance of G-CNNs is generally higher than B-CNNs, due to the fact that a discrete set of rotations is encoded into the groups of the G-CNNs. Because of that, a mismatch can happen between the rotations learned with the randomly rotated input images and the groups on the one hand, and the rotation of the images in the test set on the other hand. This mismatch is minimized for B-CNNs, as the continuous set of rotation angles is imposed by design. Thanks to that, the predictions of B-CNNs are more stable than the one of G-CNNs.

The generalized instability in performance for all methods on brain MRI (see Figure 3c) can be explained by the small number of instances used at each epoch (190 for training and 63 for testing). This characteristic of the dataset may also explain that B-CNN achieves better performance more quickly, as the rotational invariance is present by design (i.e., invariance is set as prior knowledge), by opposition to G-CNN that more heavily relies on the training instances to learn this invariance.

Figure 4 presents the results obtained on the Outex dataset in a low data setting, where 75% of the training data is removed. When, in the initial experiment presented in Figure 3b, we slightly outperformed G-CNNs by less than 5% accuracy, in this experiment where less data is provided during training, we outperform G-CNNs by a little bit less than 15%. Furthermore, the learning curves are again much more stable with B-CNNs.

## 5.5   On the equivariance of feature maps

In addition to their performance stability, B-CNNs also offer a certain stability from the equivariance of their feature maps. Unlike standard CNNs in Figure 1, the feature maps of B-CNNs extract the exact same latent features, no matter the orientation of the input image. This characteristic of B-CNNs can help in providing trust to its users. For instance, medical experts would see that the model extract the same latent features for tumorous brain images, and this no matter the orientation of the tumor.

Figure 5 shows the equivariance in B-CNNs and G-CNNs. The feature maps in the figure are randomly taken from the second convolution layer of the B-CNN and G-CNN. It can be observed that the feature maps in the first row (for a non-rotated image from Outex) for the B-CNN are generally the same as those of the second row (corresponding to the same image, but rotated of $60°$, see Figure 5a). However, when training a G-CNN with the same number of layers and the same filter sizes, its feature maps change significantly after rotating the input image (see Figure 5b).

In addition to Figure 5, we also performed a quantitative study in order to assess the rotational equivariance of B-CNNs, G-CNNs and CNNs. To do so, we performed the same experiment as the previous one, but for the whole test set and for random rotational angles. For each sample, the 4 feature maps that are the most active are min-max scaled between 0 and 1 (this step is needed since feature maps of different architectures may have absolute values at a different scale). Then, the difference between the non-rotated and the rotated feature maps is computed (the difference between the first and second row in Figure 5). The bigger this value, the less equivariant the method is. We obtained a mean value of $1.77\% \pm 2.78\%$ for the B-CNN, $10.64\% \pm 5.64\%$ for the G-CNN and $5.73\% \pm 5.64\%$ for the CNN. B-CNNs therefore achieve a better rotational equivariance.

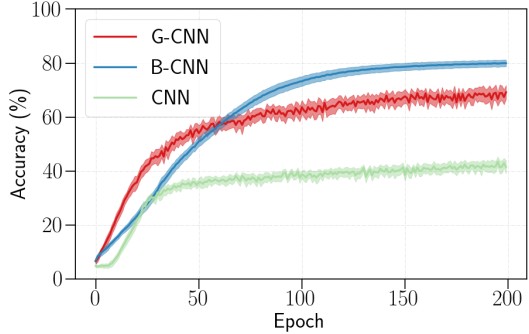

Figure 4: Performance on the Outex dataset in low data setting (120 images in the training set).

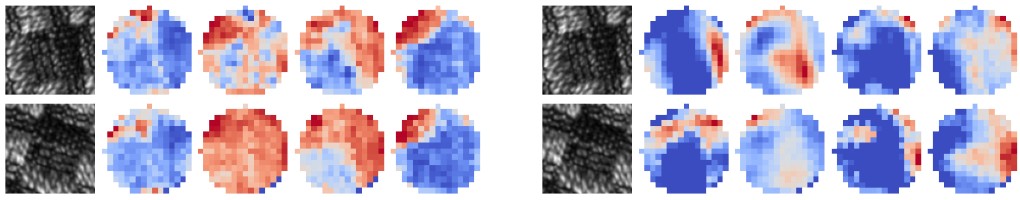

(a) Some feature maps of a B-CNN        (b) Some feature maps of a G-CNN

Figure 5: The first row corresponds to random feature maps for a non-rotated image from the Outex dataset. The second row corresponds to the same feature maps, but for a 60° rotated version of the image. To ease the comparison with the first row, these feature maps have been reoriented. The corner pixels of the images of the second row are hidden as they correspond to pixels that are out of the bound. For the sake of comparison, the same pixels have been hidden in the images of the first row.

## 6 Discussion and future work

In practice, we are not always able to obtain a perfect rotational invariance. The main reason is that the Bessel functions are discretely sampled on a grid, with a resolution fixed by the size of the filters. It directly follows that, if too small filters are used, perfect rotational invariance may not be achieved due to discretization errors. This is the reason why data augmentation helps, even when using B-CNNs. Therefore, using larger filters helps to achieve a better rotational invariance. It would be interesting to consider other interpolation methods to improve the sampling of input images and reduce this effect (similarly to what is already done by [30] to ensure translation invariance in CNNs).

In future works, one could consider bringing several updates to B-CNNs. Firstly, the actual $\nu_{max} + 1$ convolutions can be converted to only one convolution. Secondly, performances of B-CNNs are deeply dependent in this work on the choice of $\nu_{max}$ and $j_{max}$. However, it is possible to consider the Nyquist frequency in order to rather impose an upper limit to the $k_{\nu,j}$ as done in [21]. This will both minimize the aliasing effect and maximize the amount of information preserved by the Bessel coefficients. Thirdly, it is also possible to bring other types of invariance using the same framework, like for example invariance under reflections (in order to make B-CNNs E(2)-invariant).

Before concluding, we would like to point out again that B-CNNs achieve a global rotational invariance by a succession of local invariance, provided by the multiplication between the image and the filters (it is the case for all similar techniques). This property can probably make B-CNNs and similar models more easily fooled by adversarial techniques. In the future, it could be interesting to evaluate the robustness of such rotation invariant CNNs, as adversarial techniques is an important concern for many applications.

## 7 Conclusion

This paper proposes a new kind of CNNs, called Bessel CNNs (or B-CNNs), that are intrinsically invariant to the rotation of input images. In order to achieve this, a new convolutional layer has been designed, based on Bessel functions from physics. Experiments show that B-CNNs do not need any preprocessing of the input images to be rotational invariant, and that the full rotational invariance brought by the Bessel convolutions lead to more stable results. We conclude that Bessel convolutions can be used in any application involving global and/or local rotations of images.

## Acknowledgments and Disclosure of Funding

The authors want to thank José Oramas M. from the University of Antwerp for his insights on the paper. A.M. is funded by the Fund for Scientific Research (F.R.S.-FNRS) of Belgium. V.D. benefits from the support of the Walloon region with a Ph.D. grant from FRIA (F.R.S.-FNRS). This research used resources of PTCI at UNamur, supported by the F.R.S.-FNRS under the convention n. 2.5020.11.

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
