# OpenReview forum: "Achieving Rotational Invariance with Bessel-Convolutional Neural Networks"
_NeurIPS.cc/2021/Conference — NeurIPS 2021 Poster_

### Official Review · Reviewer_oaoL · 2021-07-10

**Rating:** 4
**Confidence:** 5

**Summary:**

The submission proposes Bessel-convolutional neural networks (B-CNNs), which are by design guaranteed to be equivariant w.r.t. translations and continuous rotations in two dimensions, i.e. the group SE(2). Convolution kernels are hereby expanded in an orthonormal function basis which consists of Bessel functions in the radial part and circular harmonics in the angular part (as commonly done in previous work). The central building block of B-CNNs is a non-linear operation, which (in its final form, eq. 22) computes the convolution response of the image with each angular harmonic frequency component individually, takes their square norm to produce locally rotation invariant responses, and finally aggregates those. This operation is motivated by an initial operation in eq. 12, which performs a group averaging over all filter or image rotations. Sections 4.3 and 4.5 are analytically manipulating this initial formulation in eq. 12 to its final formulation in eq. 22.

B-CNNs are evaluated on three small scale classification datasets, namely MNIST digits, Outex-TC-00010-r textures and brain MRI images. Each experiment compares B-CNNs with a non-equivariant CNN baseline and a G-CNN architecture, which is equivariant w.r.t. translations and rotations by multiples of 90 degrees. Classification accuracies (figure 3) are reported in two settings: First, each model is trained on non-rotated inputs and then tested on rotated samples. This setting evaluates the OOD generalization over rotations which is theoretically ensured by rotation equivariant networks. The second setting trains and tests on rotated samples. While the equivariant models perform significantly better than the standard CNN baseline, neither B-CNNs nor G-CNNs are clearly outperforming each other in all settings.
Section 5.5 and figure 4 present a small experiment which compares the equivariance of B-CNN, G-CNN and standard CNN feature maps. B-CNNs perform hereby slightly better than G-CNNs, which in turn perform better than CNNs. This is not surprising since the G-CNNs which are used are in contrast to the B-CNNs not continuously rotation equivariant.

**Ethical Concerns:**

There are no ethical concerns with this paper.

**Limitations And Societal Impact:**

The authors address some limitations of the model.
No immediate negative societal impact is to be expected.

**Main Review:**

Originality:
A main argument of the authors (e.g. lines 6-8 and 31-32) is that their model is the first which is equivariant w.r.t. continuous rotations in SO(2). This claim is incorrect as there exist many continuously rotation equivariant models, for instance:
- Harmonic Networks (Worral et al., CVPR 2017)
- E(2)-CNNs (Weiler and Cesa, NeurIPS 2019)
- LieConv (Finzi et al., ICML 2020)
- Zernet (Sun et al., 2018)
- Harmonic surface networks (Wiersma et al., Transactions on Graphics, 2020)

The method relies essentially on the use of a decomposition of the kernels' angular part into circular harmonics. It has been shown by (Lang and Weiler, ICLR 2021) that any SO(2)-equivariant model necessarily relies on such a decomposition (sometimes implicitly). However, the kernels' radial part is shown to be entirely unconstrained in rotation equivariant models - the use of Bessel functions to expand the radial parts is therefore arbitrary for the rotation equivariance of B-CNNs. The authors should clarify this point since the current submission gives the impression that this choice is essential (e.g. lines 8-11, 28-29). The use of a polynomial radial part is not new (see e.g. Zernet, who use Zernicke polynomials) but is in principle an interesting choice which is worth to be investigated.


Quality:
The presented theoretical results seem in general to be technically correct. The mathematical notation could at some points be made mode formal, however, this is not a big issue in an applied field. The authors may for instance choose to adapt the following points:
(line 51, 55): [0,2pi] should probably be changed to [0,2pi)
(line 51): the group action R on the input and output of an equivariant function f may in general differ
(line 104 / eq. 6): the basis is not only orthogonal but also orthonormal
(lines 101-105): this sentence is a bit confusing. Eq. 5 does not use proper set builder notation (the ", \nu, j in N" should be inside the curly brackets).

The empirical evaluation is formally correct and evaluates the key theoretical claims. However, it would be desirable to evaluate the models beyond toy datasets. The claim in line 184 that MNIST-rot consists of digits which are rotated by multiples of 2pi/8 is incorrect. A comparison against previous work on this dataset would be interesting since it is the main toy benchmark for rotation equivariant models. Such an experiment would be particularly interesting since the current submission evaluates the model only against their own weak baselines. Since a particular focus is put on continuous rotation equivariance, it would furthermore be interesting to add a harmonic network baseline and a G-CNN baseline for G=C_N with N>4.


Clarity:
The submission is mostly clearly written, however, the irrelevant role of the Bessel radial parts for rotation equivariance remains unclear (as stated above, only the circular harmonic angular parts are relevant). The necessity of the derivation in Sections 4.3 - 4.5 is in my view unclear. One could immediately propose eq. (22) as an rotationally equivariant operation without deriving it from eq. (12), which is itself not further motivated.


Significance:
Given that continuously SE(2)-equivariant models which are based on kernels with circular harmonic angular parts, the significant contributions of the submission are 1) the use a Bessel expansion of the radial parts and 2) the nonlinear mapping of non-zero frequency harmonics responses to invariant scalars via the square norm. While the current experiments evaluate the continuous equivariance of B-CNNs, they are not investigating the significance of these two choices. The authors should therefore add experiments with baselines that use 1) different radial parts and 2) different methods to map the higher order responses to scalars.

**Time Spent Reviewing:**

6

---

> ### Author Response · Authors · 2021-08-09
> **Author Response to Reviewer oaoL**
>
> Dear reviewer,
>
> We honestly want to thank you for your very interesting comments and the time you spent on our paper. We are happy to read the overall appreciation about the quality and the creativity of the paper, and we carefully took note of all your interesting remarks.
>
> As a first answer, we want to acknowledge and say that we agree about the existence of the kernels with circular harmonic angular parts. We however think that it can really be interesting for the literature to have another strong mathematical basis for developing equivariant CNNs (in our case, derived from Bessel functions). We deeply believe that the literature can rely on and build upon our Bessel-function developments to advance the field.
>
> ***“While the equivariant models perform significantly better than the standard CNN baseline, neither B-CNNs nor G-CNNs are clearly outperforming each other in all settings.”***
>
> In order to better emphasize our contribution, we ran an experiment with way smaller datasets.  This experience is in fact exactly the same as the one performed on the Outex dataset, except that we removed 75% of the training set. When, in the initial experiment presented in the paper, we slightly outperformed G-CNNs by less than 5% accuracy, in this new experiment where less data is provided during training, we outperform G-CNNs by roughly 15%. Furthermore, the learning curves are much more stable with B-CNNs. We can confidently state that B-CNNs are crucial when there is not enough data for G-CNNs to learn the equivariance. This is due to the fact that in B-CNNs, the mathematically rigorous rotation equivariance can be seen as a prior during training, and does not rely on the number of data, as it is partially the case for G-CNNs. This result will be included in the next version of the paper.
>
> ***“Section 5.5 and figure 4 present a small experiment which compares the equivariance of B-CNN, G-CNN and standard CNN feature maps. B-CNNs perform hereby slightly better than G-CNNs, which in turn perform better than CNNs. This is not surprising since the G-CNNs which are used are in contrast to the B-CNNs not continuously rotation equivariant.”***
>
> The non-continuous property of G-CNNs are in fact the issue that we want to solve with B-CNNs. Showing that B-CNNs outperform G-CNNs in the mentioned experience is therefore an empirical proof that tackling this non-continuous property matters. Of course, we can increase the number of rotations in the group of symmetry used in G-CNNs, but it will drastically increase the number of parameters. However, thank you for this comment. It would be interesting to find for a particular dataset the number of parameters needed (which is linked to the group of symmetry used) by G-CNNs to achieve a good rotational invariance, and compare it with the number of parameters required by B-CNNs. By taking advantage of the particular representation that involves Bessel functions, a good rotational invariance is easier to achieve and requires less parameters.
>
> ***“However, the kernels' radial part is shown to be entirely unconstrained in rotation equivariant models - the use of Bessel functions to expand the radial parts is therefore arbitrary for the rotation equivariance of B-CNNs. The authors should clarify this point since the current submission gives the impression that this choice is essential (e.g. lines 8-11, 28-29).”***
>
> The use of Bessel functions is justified in physics when solving wave equations. They provide a complete orthonormal basis for squared-integrable functions that we can use here to develop images in a disk. We agree with the harmonic-network approach that it is the $e^{im\phi}$ factor that makes it actually possible to obtain rotational equivariance and that the Bessel functions could be replaced by other (trainable) radial functions. We chose in this work to use Bessel functions as in physics, since we know that they form an appropriate basis to span images in a disk and because we can rely on convergence criteria to appropriately limit the number of these basis functions.
>
> ***“Quality: The presented theoretical results seem in general to be technically correct.”***
>
> Thank you for your appreciation. We will consider all the elements you provided to increase the relevance of our method.
>
> ***“The empirical evaluation is formally correct and evaluates the key theoretical claims. However, it would be desirable to evaluate the models beyond toy datasets.”***
>
> The datasets have been chosen in an increasing difficulty. MNIST is simple, but is used in almost every image analysis task as a benchmark. Outex, which is already way more complex than MNIST, is also used in different works to assess local and global rotational invariance. Then, brain MRI is used as a real-world, complex dataset, which contains local entities (the brain tumors) that can be subject to arbitrary orientations. We do not consider brain MRI as a toy dataset.
>
> ***“[...] the irrelevant role of the Bessel radial parts for rotation equivariance remains unclear (as stated above, only the circular harmonic angular parts are relevant).”***
>
> Please see our previous answer on the use of Bessel functions.
>
> ***“The necessity of the derivation in Sections 4.3 - 4.5 is in my view unclear. One could immediately propose eq. (22) as an rotationally equivariant operation without deriving it from eq. (12), which is itself not further motivated.”***
>
> In the future version of our paper, we will further motivate why we proposed Equation (12) as a starting point. In our opinion, it was easier to see and to understand the rotational invariance by looking at Equation (12) than by looking directly at Equation (22). In Equation (12), we perform an integration over $\alpha$ between $0$ and $2\pi$. As we showed in Equation (11), multiplying the Bessel coefficients by $e^{-i\nu\alpha}$ translates a rotation by an angle $\alpha$ of the initial image. Equation (12) performs a complete revolution of the image and is then automatically rotational invariant.

---

> > ### Comment · Reviewer_oaoL · 2021-08-17
> > **Reviewer response to author rebuttal**
> >
> > Dear authors,
> >
> > Thank you for your detailed response. As I see my main concerns not being sufficiently addressed, I will stick to my initial rating. In particular, the submission remains incomplete since it lacks a discussion of and benchmarking against other methods like the continuously equivariant Harmonic Networks and E(2)-CNNs or group convolutions for high order cyclic groups. Furthermore, the benefit of using Bessel functions instead of a circular harmonic basis with an alternative radial part remains unclear and would have to be demonstrated empirically.

---

### Official Review · Reviewer_gUNY · 2021-07-15

**Rating:** 8
**Confidence:** 5

**Summary:**

The paper aims at achieving rotation invariance of images by introducing Bessel transforms to deep convolutional neural networks.
Local operations are carried out on submatrices over sub-regions by shifting kernels and the inner products are taken for Bessel coefficients over radial directions. The experiments show that the proposed Bessel-convolutional neural networks perform extremely well.

**Limitations And Societal Impact:**

It would be great somewhere else if the authors could provide some theoretical justification for the proposed method. Though some computations are given on page 6 about the implementation of the Bessel transform, more details and computational complexity should be provided.

**Main Review:**

The main idea is to use Bessel transforms for representing images and to take inner products on coefficients over the radial directions. Since Bessel transforms turn rotations into multiplications with exponentials, the induced inner products becomes rotation invariant. This is a very nice observation and the authors' idea is novel. The experiments with MNIST, Outex and brain MRI provide very convincing results. The paper is also well written.

**Time Spent Reviewing:**

4

---

> ### Author Response · Authors · 2021-08-09
> **Author Response to Reviewer gUNY**
>
> Dear reviewer,
>
> Thank you very much for your time and for your positive comments.
>
> ***“Though some computations are given on page 6 about the implementation of the Bessel transform, more details and computational complexity should be provided.”***
>
> Calculating the computational complexity is not an easy task for our model, but is definitely something important when choosing the right network. We believe that the main strength of B-CNNs is its ability to deal with very small datasets (too small for other networks to learn the invariance).
>
> We ran additional experiments with smaller datasets, and it can clearly be seen that B-CNNs outperform its competitors when the data is scarce. This experience is in fact exactly the same as the one performed on the Outex dataset, except that we removed 75% of the training set. When, in the initial experiment presented in the paper, we slightly outperformed G-CNNs by less than 5% accuracy, in this new experiment where less data is provided during training, we outperform G-CNNs by roughly 15% (we will elaborate on that and show the results of the additional experiment in the final version of the paper). Furthermore, the learning curves are much more stable with B-CNNs. This means that if B-CNNs are indeed especially relevant in a low data setup, the computational complexity should be less of an issue.

---

> > ### Comment · Reviewer_gUNY · 2021-08-17
> > **Acknowledgement**
> >
> > Thanks for your response.

---

### Official Review · Reviewer_nxbX · 2021-07-16

**Rating:** 4
**Confidence:** 5

**Summary:**

The paper introduces Bessel convolutions, an approach to computing rotation equivariant representations using the well-known Bessel functions. The operation can be described mathematically as an integral over rotations of  an inner product between a rotated/phase shifted filter and the input. It is shown that this can be computed efficiently without an integral, as a modulus of a complex inner product between filter and input. The idea is validated by training networks on rotated MNIST, Outex (textures), and brain MRI scans. Results show that in among the specific models that were compared, the B-CNN outperforms on several datasets / configurations.

**Limitations And Societal Impact:**

As noted in the main review some limitations could be better discussed. Societal impact for this paper is the same as any other paper in (equivariant) CNN design, in that CNNs can be used for positive and negative uses. In practice, equivariant CNNs have mostly been applied in medical imaging and problems in the natural sciences.

**Main Review:**

The paper is well written, and technically sound. The approach is clearly motivated and explained, and enough detail is provided to enable others to reproduce the work. However, despite many strong points of the paper, I am not convinced that this method truly advances the state of the art in rotation-equivariant convolutional networks. There are a number of reasons for this:

- The method is compared to CNNs and G-CNNs that are equivariant to 90 degree rotations. It is natural to expect that a method dealing with continuous rotations can do better than 90 degree rotations, and indeed the results confirm that in several configurations (but not all) the B-CNN outperforms the G-CNNs. However, there are several methods that deal with continuous / finely discretized rotations, and these are not compared to. The main methods are harmonic networks by Worrall and, what I believe to be the current best approach, the "General E(2)-equivariant steerable CNNs" of Weiler & Cesa. Open source code is available for both, so it should not be too hard to benchmark against these.

- The networks trained are relatively small (5-7 layers) without modern design elements such as residual connections and batchnorm. Also, the datasets used for benchmarking either don't have an established state of the art result or these are not reported/compared to. For these reasons it is not clear if the comparisons reported in the paper reflect the very best that the G-CNN and B-CNN can achieve. It would be more convincing to see that B-CNNs can outperform the results reported in other papers on a competitive benchmark. There are a number such benchmarks that can be found in other papers on rotation equivariant CNNs.

- For some of the plots in figure 3, it seems like the baseline methods are not yet converged. Also, slower convergence seen for the CNN/G-CNN could indicate that the weights are not initialized at the right scale or optimizer hyperparameters are not properly tuned. Again, such concerns could be taken away by reporting results on a competitive benchmark.

- It is claimed that B-CNNs achieve equivariance to continuous rotations, but this is only guaranteed in the continuous setting, not when the Bessel functions are discretely sampled on a grid. To achieve rotation equivariance in practice, one would need to use low frequencies or relatively large filters (which have not been favored in modern CNN designs). Weiler & Cesa have an extensive discussion of this. The same is true for any steerable CNN, including ones based on circular harmonics such as harmonic networks and Weiler & Cesa's networks, so it's no shame but at least should be explained clearly.

- The property of Bessel functions that makes the B-CNN equivariant is the fact that they undergo a phase shift under rotation. That is, they are steerable filters, and so this methods falls in the category of steerable CNNs. It is not clear why the Bessel functions are a better choice than say circular harmonics, which have the same property and have been used in previous works (Worrall and Cesa & Weiler). The circular harmonics already provide a complete basis for the space of equivariant filters, which implies one does not need anything else. Perhaps there is a numerical reason why Bessel functions are more stable or otherwise preferable, but this should then be explained clearly and demonstrated empirically in a head-to-head comparison with circular harmonics based approaches.

- I'm not 100% clear on the implementation details, but it seems that the current implementation requires nu_max+1 calls to conv2d, which would make it considerably slower than methods that use a single call. Perhaps the implementation can be improved though?

Although I have focussed on points that need to be improved, I want to reiterate that this is not a bad paper at all. In fact it's based on a neat idea, with solid and interesting underlying mathematics. However to really advance the field it is necessary to deeply study the current best methods, understand and address their weaknesses. Studying steerable filter bases for equivariant CNNs is a topic worthy of study, so I encourage the authors to dig deeper into this topic to better understand what makes for a good basis of filters, and perform direct comparisons to the most closely related and best performing existing methods.

**Time Spent Reviewing:**

3

---

> ### Author Response · Authors · 2021-08-09
> **Author Response to Reviewer nxbX**
>
> Dear reviewer,
>
> Thank you for your time and your very positive comments. It is always a pleasure to read that “The paper is well written, and technically sound. The approach is clearly motivated and explained, and enough detail is provided to enable others to reproduce the work.” We noted all your remarks, notably on harmonic networks, to improve the next version of our paper.
>
> ***“However, despite many strong points of the paper, I am not convinced that this method truly advances the state of the art in rotation-equivariant convolutional networks.”***
>
> We want to stress what we think are the most interesting parts of our paper. First, we propose a mathematically strong and novel approach to rotational invariance in CNNs, and we deeply believe that the literature can build on this basis to improve the field. Second, our experiments show that the way we learn rotational invariance provides more stable results. To elaborate more on this point, we ran additional experiments in a small data setting, where it can clearly be seen that B-CNNs outperform its competitor (as it relies way less on the size of data to learn the invariance). This experience is in fact exactly the same as the one performed on the Outex dataset, except that we removed 75% of the training set. When, in the initial experiment presented in the paper, we slightly outperformed G-CNNs by less than 5% accuracy, in this new experiment where less data is provided during training, we outperform G-CNNs by roughly 15%. Furthermore, the learning curves are much more stable with B-CNNs. We will add more discussions / experiments about that in the future version of the paper.

---

> > ### Comment · Reviewer_nxbX · 2021-08-16
> > **Remaining issues**
> >
> > Dear authors,
> >
> > Thanks for the response to my review. Although you raise some good points in your rebuttal, my main concerns still stand: the method presented has not been compared to the state of the art in continuous rotation equivariant CNNs (i.e. general E(2) equivariant steerable CNNs), and at the same time the method is almost certainly slower, requiring many calls to conv2d whereas steerable CNNs require only one call to implement the equivariant convolution. I will thus keep my rating as is.

---

### Official Review · Reviewer_dNyU · 2021-07-16

**Rating:** 5
**Confidence:** 4

**Summary:**

This work proposes a rotationally invariant CNN based on local Fourier-Bessel expansion of input images. To define the new type of convolution, for each local region of the image, the proposed B-CNN takes the averaged (over angles) inner product between kernel and “rotated” Fourier-Bessel coefficients. This convolution is invariant to the rotation of the local patches. Unlike G-CNN that deals with discrete sets of rotations, the B-CNN can handle any rotation in the continuous space $[0, 2\pi]$. The numerical experiments demonstrate the advantage of B-CNN in various settings.


**Limitations And Societal Impact:**

Yes

**Main Review:**

The paper is in general well-written. In theory, it is a nice property that B-CNN is able to achieve rotation invariance for any continuous rotations. It seems to perform better than G-CNN especially when there is no data argumentation. However, I have the following concerns on the novelty and experiments of the paper:

1). I feel that the contributions of this paper are over claimed. For example, the paper claims itself to be the first work in machine learning that introduces the Bessel functions. However, the following papers already introduced Fourier-Bessel, and in general, steerable basis functions, to achieve rotation invariance in dimension reduction, image classification/recognition/denoising:

1. “Fast Steerable Principal Component Analysis”.
2. “Steerable PCA for Rotation-Invariant Image Recognition”.
3. “Fourier–Bessel rotational invariant eigenimages”.
4. “Rotationally invariant image representation for viewing direction classification in cryo-EM”.

The motivation of the above works is almost the same as this paper. For example, see (15)-(20) in https://www.ncbi.nlm.nih.gov/pmc/articles/PMC3711886/ that uses similar integral over angle $\alpha$ compared to (12) of this B-CNN paper. The only difference is that (15)-(20) in the above link computes the rotationally invariant (angular averaged) mean (or the inner product with a constant kernel) and covariance, and in this B-CNN paper (12) computes the rotationally invariant (angular averaged) inner product with a kernel (not a constant kernel). I hope the authors can clarify the strong relevance of these papers.

2). The experimental setup looks a bit unfair for G-CNN which uses $\mathbb Z_4$ to represent the discrete rotations. I understand that the original G-CNN paper also uses $\mathbb Z_4$ group, but it seems to be generalizable to e.g. $\mathbb Z_8$, $\mathbb Z_{16}$ (although careful interpolation may be required), at the cost of increasing the number of parameters. However, since in the experiment B-CNN has fewer layers than G-CNN, one can certainly reduce the number of layers in G-CNN and increase its number of discrete rotations to keep the similar number of model parameters. Another simpler way is to consider implementing CNN on the polar-transformed image with cyclic paddings, like in paper “Polar Transformer Networks”. I am curious about the experimental results in this new setting.

3). The paper claims its advantage on handling continuous rotations, but I doubt its importance for discrete images. For example, for images with low resolution, due to the discretization error it is sufficient to use the set of discrete rotations. The number of rotations $n_R$ should be proportional to the image resolution. One may argue that for high resolution more discrete rotations are needed and this increases the model complexity of G-CNN. However, the same issue applies to B-CNN. Indeed, for low resolution images, one should use smaller $\nu_{\max}$ and $j_{\max}$ to avoid the aliasing problem. Namely, one should keep them below the Nyquist frequency of the image, which is proportional to the image resolution. For higher resolution, one should increase the number of angular and radial frequencies accordingly and this increases the model complexity. From this perspective, the parameter $\nu_{\max}$ in B-CNN and $n_R$ in G-CNN play similar roles (although the two are respectively in frequency and real domain). Thus, it is a bit suspicious that you take a large $\nu_{\max}$ in B-CNN but a much smaller number of discrete rotations ($n_R=4$) in G-CNN (see also my point 2) ). The question is that when $\nu_{\max} \approx n_R$ and with the same number of layers, can B-CNN still beat G-CNN? A more convincing experiment set up is to compare the performance between dfifferent neural networks with the same number of layers for varying $\nu_{\max}$ and $n_R$ .

Minor comments:
1. In (5), $\nu', j’$ should be $\nu, j$.
2. In line 105, $J’$ should be $J$.
3. In the first line below line 143, the subscript of the last $\theta$ should be $\nu', j’$.

Missing references:
1. Dense Steerable Filter CNNs for Exploiting Rotational Symmetry in Histology Images.
2. General E(2) - Equivariant Steerable CNNs.


In summary, this paper applies ideas from Fourier-Bessel steerable basis to CNNs, and is well-written in general. However, its novelty is limited in my opinion (which is overclaimed in the paper), and more experiments and clarifications are expected.


**Time Spent Reviewing:**

12 hrs

---

> ### Author Response · Authors · 2021-08-09
> **Author Response to Reviewer dNyU**
>
> Dear Reviewer,
>
> Thank you for your comments and your time that you spent on our paper. Your comments are really interesting and will be taken into account for sure in the next version of our paper.
>
> Regarding your note about our B-CNN that ***“its novelty is limited in my opinion”***, we believe that we provide a new strong mathematical basis for rotational-invariant CNNs that can be easily exploited and improved by the literature. We show that our method can compete with G-CNNs and that it is more stable.
>
> New experiments that we ran also show that we clearly outperform G-CNNs in a low data setup, where our prior inside the network makes the learning of the invariance much easier. This experience is in fact exactly the same as the one performed on the Outex dataset, except that we removed 75% of the training set. When, in the initial experiment presented in the paper, we slightly outperformed G-CNNs by less than 5% accuracy, in this new experiment where less data is provided during training, we outperform G-CNNs by roughly 15%. Furthermore, the learning curves are much more stable with B-CNNs. We will add this new information in the next version of our paper.
>
> ***“I hope the authors can clarify the strong relevance of these papers.”***
>
> Thank you for pointing out these papers. We will discuss them in the next version of our paper.
>
> ***“However, since in the experiment B-CNN has fewer layers than G-CNN, one can certainly reduce the number of layers in G-CNN and increase its number of discrete rotations to keep the similar number of model parameters.”***
>
> We agree with this way of comparing B-CNNs with G-CNNs. However, our point of view is that showing that B-CNNs outperform G-CNNs with fewer layers than the number introduced in other papers could also have been described as unfair.
>
> ***“Thus, it is a bit suspicious that you take a large νmax in B-CNN but a much smaller number of discrete rotations (nR=4) in G-CNN. The question is that when νmax≈nR and with the same number of layers, can B-CNN still beat G-CNN?”***
>
> It would be interesting to perform an experiment similar to what you propose. Of course we can increase the number of rotations in the group of symmetry used in G-CNNs, but it will drastically increase the number of parameters. We expect that for νmax≈nR, the number of parameters required by G-CNNs will be much bigger than for B-CNNs.

---

> > ### Comment · Reviewer_dNyU · 2021-08-17
> > **Response to the authors**
> >
> > Dear authors,
> >
> > Thanks for your response. After reading your reply and reviews from other reviewers, I think this paper could be largely improved by a major revision (on both literature reviews and experiments). Thus, I prefer not to change my score. I also agree with other reviewers on the issues of lacking comparison with other related methods e.g. E(2)-CNNs. I would also like to point out that Bessel functions have already been popular in rotation-invariant dimension reduction for years, and it is not the only common steerable basis. For example, one may consider Prolate spheroidal wave functions (PSWF) which is optimal in some sense when expanding a function in a compact domain to a band-limited function. In this sense, I would like to encourage the authors to explore more on the effects of different steerable bases.

---

### Decision · Program_Chairs · 2021-09-28

**Decision:**

Accept (Poster)

**Comment:**

While reviewers enjoyed aspects of the paper, the key remaining concern after the discussion period was a distinct lack of comparisons (and theoretical advantages) over the state-of-the-art in rotation invariant neural networks. Several reviewers have left their final comments as replies to the replies. Please consider these carefully in preparing your revisions.

**Consistency Experiment:**

NeurIPS has a long history of experimentation. In 2014, NeurIPS ran an experiment in which 10% of submissions were reviewed by two independent committees to quantify the randomness in the review process. This year, we repeated a variant of this experiment to see how the quality of the review process has changed over time.  This paper was part of the experiment and was therefore assigned to two committees (consisting of reviewers, an Area Chair, and a Senior Area Chair) that reached independent decisions.  If both committees made the same recommendation, this recommendation was followed. If a single committee recommended acceptance, the paper was accepted (with the exception of a few cases in which the other committee identified what we considered a fatal flaw, e.g., an error in a key result).

This copy’s committee reached the following decision: **Reject**

The other committee assigned to the paper recommended **Accept (Poster)**.  You can find the other set of reviews, along with any follow up discussion with the authors here:
https://openreview.net/forum?id=KYrenOCQuM